# MedKamba: A Novel Approach Integrating State-Space Models and Fractional Kolmogorov–Arnold Networks for Medical Image Segmentation

**Amit Shakya** (ID)               AMITSHAKYA@YAMAHA-MOTOR-INDIA.COM
**Akanksha Yadav**                AKANKSHA@YAMAHA-MOTOR-INDIA.COM
**Rupesh Kumar**             RUPESHKUMAR@YAMAHA-MOTOR-INDIA.COM
**Lalit Sharma**                LSHARMA@YAMAHA-MOTOR-INDIA.COM
*Emerging Technologies and Innovation Lab, Yamaha Motor Solutions India*

**Editors:** Accepted for Publication at MIDL 2026

## Abstract

Medical image segmentation plays a crucial role in healthcare, serving as a key step in disease diagnosis and treatment planning. Convolutional neural networks (CNNs) are limited by their restricted receptive fields, whereas Transformer-based models suffer from quadratic computational cost. Recent advances such as Mamba, a selective state-space model with linear complexity, and its vision-oriented variant, the Visual State Space (VSS) models, have shown strong ability to capture long-range dependencies efficiently. However, they still exhibit shortcomings in segmentation tasks, including loss of pixel-level structural information and inefficient channel utilization. To address this, we introduce VSSM-based Local Aware Channel Enhancement (LACE) block, which incorporates local enhancement and channel attention to better preserve spatial detail. To this end, we proposed MedKamba, a novel U-shaped segmentation approach that employs a hybrid encoder with CNNs and LACE blocks to effectively capture both local and global contextual information. While the U-Net backbone remains highly efficient, its traditional skip connections rely on simple scale-matched fusion, limiting cross-scale interaction. To overcome this, we redesign the skip connections using Fractional Kolmogorov–Arnold Networks ($f$-KANs) to generate channel-wise attention weights from features aggregated across multiple stages. Experiments on two benchmark datasets demonstrate that MedKamba consistently outperforms competing approaches and produces more visually accurate segmentation results. Code is available at https://github.com/amit-shakya-28/MedKamba-Medical-Image-Segmentation-using-KAN-and-Mamba

**Keywords:** Medical Image Segmentation, State Space Models, Fractional KANs.

## 1. Introduction

Over the past decade, medical image segmentation has become an indispensable component of computer-aided diagnosis and surgical planning. It enables precise delineation of anatomical structures and lesions, thereby improving diagnostic accuracy and treatment efficiency. Traditional segmentation methods using handcrafted features and thresholds often struggle with the complexity and variability of medical data (Dai et al., 2024).

With the advancement of deep learning, CNNs and Transformer models have greatly improved medical image segmentation, with CNNs like U-Net (Ronneberger et al., 2015), ResNet (Targ et al., 2016), nnU-Net (Isensee et al., 2021), and SegResNet (Tang et al., 2023)

effectively capturing local structures via shared kernels and pooling. However, CNNs are limited by their local receptive fields, making it difficult for them to capture global contextual information. Despite the theoretical expansion offered by dilation or deeper stacking, the effective receptive field achieved in real models is significantly smaller. Moving beyond convolutional constraints, Transformer-based architectures such as Swin Transformer (Liu et al., 2021), UNETR (Hatamizadeh et al., 2022), and SwinUNETR (Hatamizadeh et al., 2022), leverage multi-head self-attention (MHSA) to model long-range relationships, enabling them to effectively capture global contextual information within medical images. However, computing attention between all token pairs causes quadratic computational and memory cost, which is challenging for high-resolution medical images. Although expressive, these models often overfit small medical datasets and need large amounts of data.

Recent advances in Structured State-Space Models (SSMs) have enabled highly efficient sequence modeling for vision tasks. Notably, Mamba (Gu and Dao, 2023), a newly introduced SSM, which is notable for its hardware-efficient design and selective scanning, enabling robust global feature modeling with linear complexity. These strengths make Mamba a strong, efficient alternative to Transformers, with studies showing comparable performance across NLP and medical imaging tasks. For example, U-Mamba (?) incorporated a hybrid module into the nnU-Net framework (Isensee et al., 2021), combining the local feature modeling strengths of CNNs with Mamba's global representation capability, representing an early attempt to integrate Mamba blocks into medical image analysis. Building on Mamba, the Visual State Space (VSS) model (Liu et al., 2024c) incorporated a 2D Selective Scanning (SS2D) mechanism that scans images in four directions, enabling the extraction of richer and more diverse contextual cues. Swin-UMamba (Liu et al., 2024a) advanced earlier Mamba-based frameworks by integrating ImageNet-pretrained features, effectively merging broad visual priors with Mamba's efficient global representation learning.

However, because Mamba is inherently designed for 1D sequential inputs, applying it directly to 2D images requires flattening spatial features, which disrupts neighborhood continuity and causes local pixel forgetting. Moreover, capturing long-range dependencies demands a large number of hidden states, introducing substantial channel redundancy and weakening discriminative channel learning. To mitigate these issues, we introduce the Local Aware Channel Enhanced (LACE) block, which augments Mamba with a local convolutional module to restore spatial locality lost during flattening. Additionally, a channel-attention mechanism suppresses redundant channels that arise from large hidden-state dimensions, while a learnable scaling factor modulates the skip connection for improved feature calibration. Many recent studies highlight the importance of incorporating multi-scale and multi-stage information in medical image segmentation. For instance, (Ruan et al., 2022) proposed MALUNet, a lightweight U-shaped network that integrates four specialized modules. To enhance cross-stage feature interaction, MALUNet employs the Channel Attention Bridge (CAB) and Spatial Attention Bridge (SAB), which generate channel and spatial attention maps, respectively. These modules enable significant channel reduction while preserving strong segmentation performance. However, the MLP-based CAB still suffers from limited interpretability and quadratic complexity in feature dimensions.

To overcome all the limitations discussed above, we propose MedKamba, a U-shaped architecture whose encoders combine CNNs operations at starting stage followed by our VSSM based LACE blocks, enabling robust local–global feature modeling. Meanwhile, the

skip connections are enhanced using a KAN-based Spatial–Channel Attention ($f$-KSCA) module, providing interpretable and efficient cross-stage feature refinement. Furthermore, our approach incorporates Fractional KANs ($f$-KAN) with a Fractional Jacobi Activation Function (FJAF), a learnable polynomial-inspired activation with tunable fractional order. Unlike fixed polynomial bases (e.g. Chebyshev or Legendre), FJAF adapts its functional shape during training, offering more expressive nonlinear modeling and improved training stability (Aghaei, 2025). *To the best of the authors' knowledge, this is one of the first efforts that integrates State Space Models and fractional KANs for medical image segmentation.*

In summary, this paper makes the following contributions:

- We propose MedKamba, a U-shaped medical image segmentation model that combines a State Space Model with Fractional KAN, achieving an effective balance between accuracy and computational efficiency.

- We develop the Local-aware Channel Enhancement (LACE) block, which enhances Mamba's capability by restoring local spatial details and suppressing redundant channels.

- We present a Fractional Kolmogorov–Arnold Spatial-channel attention block ($f$-KSCA) that integrates multi-stage global contextual information during decoding while improving multi-scale local features. $f$-KSCA enables the block to model global relationships and assign context-sensitive weights to feature maps.

- Our method demonstrates strong performance improvements on two distinct medical imaging datasets, ISIC 2018 for skin lesions segmentation and BUSI for breast ultrasound, highlighting its ability to generalize across varied anatomical structures and imaging characteristics.

## 2. Methodology

**Overview**: Figure 1 shows our overall architecture of the proposed MedKamba, comprising an encoder, a decoder, and a skip connection. The encoder includes three convolutional blocks along with two newly introduced LACE block while the decoder applies two LACE blocks first and ends with 3 convolutional blocks. The encoder gradually compresses feature maps by a factor of two at each stage, while the decoder symmetrically restores their size. The number of channels at each stage, denoted as C1 to C5, is configurable; in our experiments, we use 16, 32, 128, 160, and 256 channels respectively. The skip connection is enhanced by the use of $f$-KSCA block. In the following section, we describe each component in detail.

### 2.1. Feature Modeling in MedKamba

An input image $I$ with shape $B \times C \times H \times W$ (representing batch size, channels, height, and width) is processed by the model using three main stages: an encoder that extracts features, skip connections that preserve important information, and a decoder that reconstructs the final output. Within the encoder, the image first goes through a convolutional

block that produces the feature map $x_{e,1} \in \mathbb{R}^{B \times 16 \times \frac{H}{2} \times \frac{W}{2}}$. It then flows through two additional convolutional modules, generating higher-level features $x_{e,2} \in \mathbb{R}^{B \times 32 \times \frac{H}{4} \times \frac{W}{4}}$ and $x_{e,3} \in \mathbb{R}^{B \times 128 \times \frac{H}{8} \times \frac{W}{8}}$. Each module applies two $3 \times 3$ convolutions followed by a max-pooling layer to downsample the spatial resolution. Here, $x_{e,i}$ denotes the feature map generated at the $i$-th encoder level. The feature map $x_{e,3}$ is first downsampled using a Patch Embedding layer, implemented as a $2 \times 2$ convolution with stride 2, which decreases the spatial dimensions while expanding the channel depth. The output of this operation is then passed through VSSM based LACE block, producing the feature map $x_{e,4} \in \mathbb{R}^{B \times 160 \times \frac{H}{16} \times \frac{W}{16}}$. Section 2.2 outlines the LACE block in details. The skip-connection pathway employs the $f$-KSCA to refine the feature maps $x_{e,1}$, $x_{e,2}$, $x_{e,3}$ and $x_{e,4}$ from the first four encoder stages by emphasizing the most informative channel and spatial responses. The enhanced outputs, denoted as $s_1$, $s_2$, $s_3$, $s_4$, are then passed to the decoder for feature fusion. Section 2.3 provides a comprehensive description of the $f$-KSCA module. The decoder reconstructs the spatial details in a manner that parallels the encoder. We begin by upsampling the feature map $x_{e,5}$ using interpolation, where a linear layer increases the channel dimension and a rearrangement converts it to $\mathbb{R}^{B \times \frac{H}{16} \times \frac{W}{16} \times 256}$. This representation is fused with the skip-connected feature $x_{e,4}$ by channel concatenation. Repeating the same upsampling procedure generates $x_{d,4} \in \mathbb{R}^{B \times \frac{H}{8} \times \frac{W}{8} \times 160}$. Subsequently, three convolutional stages progressively increase the spatial resolution and refine the decoded features. A final $1 \times 1$ convolution compresses the channel dimension and generates the segmentation output.

## 2.2. Local Aware Channel Enhancement (LACE) block

Earlier Transformer-based segmentation models generally employ a block structure that follows the sequence: *Norm–Attention–Norm–MLP.* Although both SSMs and Attention can capture global dependencies, they exhibit different behaviors, and thus directly substituting Attention with SSM leads to sub-optimal performance, this suggests that designing a new block specifically for segmentation models could be highly effective.

To this end, we introduce the Local Aware Channel Enhancement (LACE) to tailor the SSM module for medical image segmentation. As shown in Fig. 1(a), for an input deep feature map $F^D \in \mathbb{R}^{B \times C \times H \times W}$, the processing starts with a LayerNorm (LN) operation, and then uses the Vision State-Space Module (VSSM) (Liu et al., 2024c) to model long-range spatial dependencies. We incorporate a learnable scale factor $s \in \mathbb{R}^C$ to modulate skip-connection features.

$$V^l = \text{VSSM}(\text{LN}(F^D)) + s \cdot F^D \tag{1}$$

Since, SSMs handle feature maps as one-dimensional token sequences, the chosen flattening strategy plays a major role in determining how many neighboring pixels remain adjacent in the sequence. When using the four-direction unfolding strategy of (Liu et al., 2024b), each anchor pixel retains only four immediate neighbors in the 1D token sequence. Consequently, flattening the 2D feature map places many adjacent pixels far from each other, which may lead to the loss of important local pixel information. To regain neighborhood information, we apply a local convolution after the VSSM. Specifically, we first apply LayerNorm to $V^l$ and then use convolutional layers to restore the information of the local features. To remain computationally efficient, the convolution layer employs a bottleneck design: first,

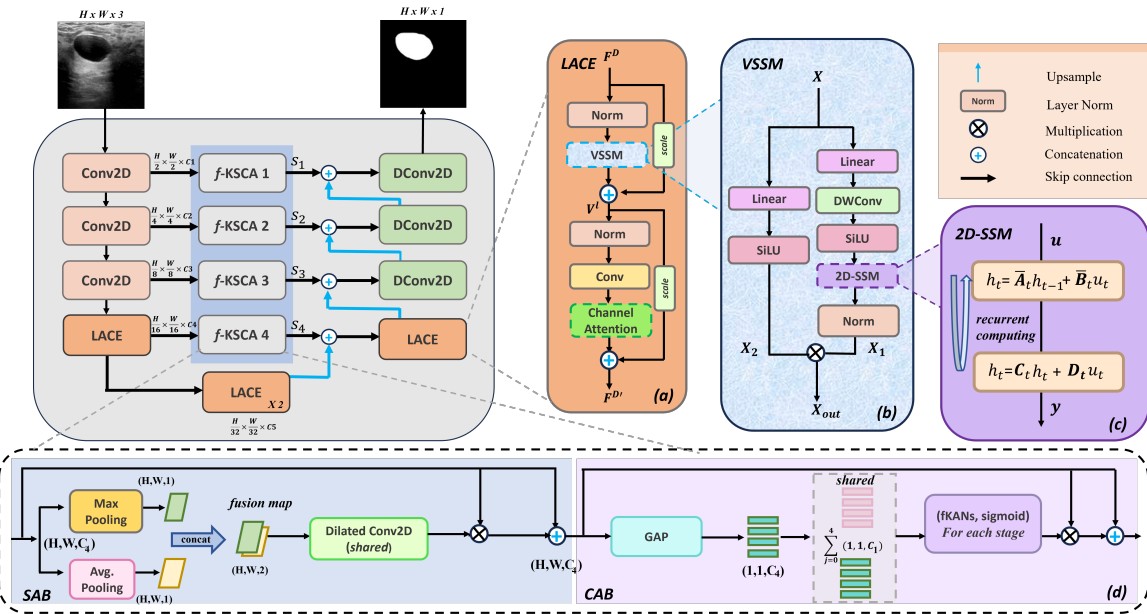

Figure 1: Overview of the proposed architecture.

the channel dimension is reduced by a factor $\beta$, producing features of size $\mathbb{R}^{H \times W \times \frac{C}{\beta}}$, and then resized to match the original number of channels. Due to the reliance of SSMs on multiple hidden states for modeling long-range interactions, they often introduce redundant channel activations, as illustrated in Fig.1(a). To alleviate this redundancy, we integrate Channel Attention (CA) (Hu et al., 2018) into the LACE block, enabling the SSM to learn diverse channel features while CA selects the most informative ones. Finally, a learnable scale factor $s' \in \mathbb{R}^C$ is added through the residual pathway to obtain the final output

$$F^{D'} = \text{CA}(\text{Conv}(\text{LN}(V^l))) + s' \cdot V^l. \tag{2}$$

### 2.2.1. VISION STATE-SPACE MODULE

Transformer-based segmentation networks frequently divide the input into tiny patches (Chen et al., 2021) or use shifted window attention (Liang et al., 2021), which restricts global interaction throughout the image, in order to maintain computational efficiency. As shown in Fig. 1(b), the Vision State-Space Module (VSSM) uses a state-space formulation to model long-range dependencies. In accordance with (Liu et al., 2024b), two parallel branches process the input feature $X \in \mathbb{R}^{H \times W \times C}$. The first branch uses a linear layer to increase the channel dimension to $\lambda C$. This is followed by a 2D-SSM layer, LayerNorm, depth-wise convolution, and SiLU activation (Shazeer, 2020). Additionally, the second branch uses SiLU activation and projects the channels to $\lambda C$. Lastly, a Hadamard (element-wise) product is employed to fuse the outputs from both branches.

The element-wise product of the two branches is projected back to $C$ channels to form $X_{\text{out}}$:

$$X_1 = \text{LN}\big(2\text{DSSM}(\text{SiLU}(\text{DWConv}(\text{Linear}(X))))\big)$$
$$X_2 = \text{SiLU}(\text{Linear}(X)) \tag{3}$$
$$X_{\text{out}} = \text{Linear}\big(X_1 \odot X_2\big)$$

where DWConv denotes depth-wise convolution and $\odot$ represents the Hadamard product.

**2D Selective Scan Module** The traditional Mamba (Gu and Dao, 2023) can only access data from the section of the sequence that has previously been scanned since it handles inputs in a causal fashion. This behaviour presents challenges when applied to non-causal data, like images, even though it is suitable for NLP tasks. We use the 2D Selective Scan Module (2D-SSM) in accordance with (Liu et al., 2024b) to better utilise 2D spatial structure. The 2D feature map is flattened into four 1D sequences by scanning in four directions: top-left to bottom-right, bottom-right to top-left, top-right to bottom-left, and bottom-left to top-right. The discrete state-space equation is used to process each sequence in order to identify long-range dependencies. The original 2D layout is then restored by summing and reshaping the outputs.

### 2.3. $f$-KSCA Module

This study integrates Kolmogorov–Arnold Networks (KANs) into the U-Net framework, leveraging their high efficiency and interpretability. Unlike standard MLPs, which rely on weight matrices $W$ and fixed activations $\sigma$:

$$\text{MLP}(Z) = (W_{K-1} \circ \sigma \circ \cdots \circ W_0)Z, \tag{4}$$

KANs replace weights with learnable activation functions $\varphi_{q,p}$, forming each layer as:

$$\text{KAN}(Z) = (\Phi_{K-1} \circ \cdots \circ \Phi_0)Z, \quad \Phi = \{\varphi_{q,p}\}. \tag{5}$$

KANs are based on the Kolmogorov-Arnold representation theorem, which states that any multivariate continuous function can be expressed as a finite sum of univariate functions composed with addition (Liu et al., 2024d).

For $d$-dimensional input $\zeta$, a KAN can be formulated as:

$$\hat{\chi}(\zeta) = \sum_{q=0}^{2d} \Phi_q\Big[\sum_{p=1}^{d} \varphi_{q,p}(\zeta_p)\Big], \tag{6}$$

where $\varphi_{q,p} : [0,1] \to \mathbb{R}$ and $\Phi_q : \mathbb{R} \to \mathbb{R}$ are trainable univariate functions. Variants such as Fourier KANs (Zhang et al., 2025), Wavelet KANs (Bozorgasl and Chen, 2024) and Chebyshev KANs (SS et al., 2024) replace B-splines with orthogonal polynomials or other basis functions to reduce computational cost and improve accuracy. Despite their efficiency, polynomial-based KANs can still suffer from limited flexibility and smoothness, particularly when approximating non-polynomial or mixed-frequency signals. Although orthogonal polynomials (e.g.Chebyshev) improve numerical stability and mitigate Runge-type oscillations, they still require careful degree selection. To address these limitations, fractional Jacobi functions (Aghaei, 2025) are utilized as trainable activations in the fractional

KAN (fKAN). By introducing a learnable fractional order parameter $\gamma$ the basis functions extend classical polynomials into a continuous fractional functional space, enabling smooth interpolation between integer orders and more adaptive function approximation while preserving boundedness and closed-form derivatives.

The fKAN output is computed as:

$$\hat{\chi}(\zeta) = \sum_{q=0}^{Q} \Phi_q^{(\gamma)} \sum_{p=1}^{d} \varphi_{q,p}^{(\gamma)}(\zeta_p), \tag{7}$$

allowing the network to explore a fractional polynomial space, improve approximation flexibility, and maintain interpretability while reducing computational complexity compared to standard MLPs.

Our $f$-KSCA module integrates multi-scale, multi-stage global context, as illustrated in Fig.1(d). Unlike MALUNet (Ruan et al., 2022), we substitute the MLP with KAN. In the four-stage $f$-KSCA module, the process starts with the Spatial Attention Bridge (SAB), allowing the network to highlight key spatial details and ignore less relevant regions. To achieve this, we first apply max pooling and average pooling to the feature map, then combine the resulting maps by concatenating them. After that, a shared dilated convolution is used to merge these features, and a sigmoid function is applied to produce the spatial attention map. Finally, we multiply the spatial attention map element-wise with the original image, then add the residual information to this result to obtain the final spatial attention output. The novelty of $f$-KSCA lies in its first-time incorporation of Fractional KAN into the Channel Attention Bridge (CAB) to enhance interpretability. In this approach, Fractional KANs are used to generate channel attention maps, which then guide the fusion of residual features in later layers through adaptive weight assignment. As a result, CAB improves multi-stage features by highlighting the most important channels and suppressing those with weak contextual relevance. This process is described by the following equations.

$$\begin{aligned}
Z &= \text{Concat}(\text{AvgPool}(x_i)), \quad i \in \{1, 2, 3, 4\} \\
Att_4 &= \sigma(\text{KANs}(\text{Conv1D}(y))), \quad i \in \{1, 2, 3, 4\} \\
\text{Out}_4 &= x_4 + Att_4 \cdot x_4
\end{aligned} \tag{8}$$

AvgPool denotes global average pooling, while Conv1D represents a one-dimensional convolution operation. By leveraging the strong function-approximation capability of KAN, the proposed $f$-KSCA module efficiently combines multi-scale and multi-stage features during the decoding process, ultimately resulting in improved segmentation performance.

## 3. Experiments and Result Analysis

**Dataset** The ISIC 2018 dataset (Codella et al., 2019) is a widely adopted benchmark for skin lesion segmentation, comprising 3,694 dermoscopic images divided into 2,594 training samples, 100 validation samples, and 1,000 test samples. The images exhibit substantial variability in resolution (0.5–29 MP) and spatial dimensions, ranging from 540×576 up to 4499×6748 pixels, making the dataset a robust and diverse testbed for evaluating segmentation models. The BUSI dataset (Al-Dhabyani et al., 2020) provides breast ultrasound images with corresponding segmentation masks for normal, benign, and malignant cases.

| Model | Param. | GFlops | Spec. | Accu. | mIoU | Dice |
|---|---|---|---|---|---|---|
| Unet(Ronneberger et al., 2015) | 14.75 | 25.19 | 89.63 | 90.60 | 75.21 | 84.30 |
| MultiResUnet(Ibtehaz and Rahman, 2020) | 7.25 | 18.76 | 90.26 | 91.30 | 76.76 | 85.65 |
| nnUNet (Isensee et al., 2021) | 19.1 | 412.7 | 94.32 | 92.21 | 77.03 | 86.43 |
| U-NeXt(Valanarasu and Patel, 2022) | **1.47** | **0.57** | 91.01 | 91.33 | 77.57 | 85.79 |
| Swin-Unet(Cao et al., 2022) | 27.15 | 5.91 | 89.89 | 91.42 | 76.44 | 85.54 |
| ACC-Unet (Ibtehaz and Kihara, 2023) | 16.77 | 45.33 | 92.51 | 92.51 | 79.12 | 87.71 |
| MedSA (Wu et al., 2023) | 104.3 | 52.2 | 91.09 | 92.04 | 77.71 | 86.36 |
| VM-UNet (Ruan et al., 2024) | 27.43 | 4.11 | 95.00 | 92.18 | 75.25 | 85.88 |
| U-KAN(Li et al., 2025) | 9.38 | 6.89 | 90.05 | 92.07 | 78.20 | 86.41 |
| H-vmunet (Wu et al., 2025) | 6.43 | 1.48 | 95.01 | 92.54 | 76.38 | 86.61 |
| **MedKamba (Ours)** | 5.50 | 2.16 | **95.15** | **93.00** | **79.63** | **88.24** |

Table 1: Quantitative comparison against state-of-the-art segmentation models on ISIC 2018 dataset. The **best** results are in **bold** and second best are underlined.

| Model | Param. | GFlops | Spec. | Accu. | mIoU | Dice |
|---|---|---|---|---|---|---|
| Unet(Ronneberger et al., 2015) | 14.75 | 25.19 | 97.94 | 96.07 | 68.30 | 76.80 |
| MultiResUNet(Ibtehaz and Rahman, 2020) | 7.25 | 18.76 | 96.31 | 95.10 | 64.99 | 74.66 |
| nnUNet (Isensee et al., 2021) | 19.1 | 412.7 | 98.39 | 96.51 | 70.02 | 79.45 |
| U-NeXt(Valanarasu and Patel, 2022) | **1.47** | **0.57** | 98.09 | 96.41 | 70.13 | 79.04 |
| Swin-Unet(Cao et al., 2022) | 27.15 | 5.91 | 96.90 | 96.07 | 70.10 | 78.77 |
| ACC-Unet (Ibtehaz and Kihara, 2023) | 16.77 | 45.33 | 95.72 | 95.00 | 70.16 | 79.24 |
| MedSA (Wu et al., 2023) | 104.3 | 52.2 | 97.53 | 95.67 | 67.29 | 77.16 |
| VM-UNet (Ruan et al., 2024) | 27.43 | 4.11 | 97.59 | 95.58 | 61.36 | 76.05 |
| U-KAN(Li et al., 2025) | 9.38 | 6.89 | 97.24 | 95.90 | 70.12 | 80.01 |
| H-vmunet (Wu et al., 2025) | 6.43 | 1.48 | 98.19 | 96.36 | 66.55 | 79.92 |
| **MedKamba (Ours)** | 5.50 | 2.16 | **98.54** | **96.62** | **71.07** | **82.17** |

Table 2: Quantitative comparison against state-of-the-art segmentation models on the BUSI dataset. The **best** results are in **bold**, and the second-best are underlined.

For our experiments, we use 647 images covering benign and malignant lesions, adopting an 80:20 random split. Due to tumor variability and ultrasound noise, BUSI is a challenging benchmark for accurate breast abnormality detection and segmentation.

**Training and Implementation detail** All images were resized to $256 \times 256$, and the model was trained for 400 epochs using Adam (lr=$1 \times 10^{-4}$, momentum=0.9, weight decay=$1 \times 10^{-4}$). For KAN components, a higher lr of $1 \times 10^{-2}$ with the same weight decay was applied to stabilize spline learning. All experiments were implemented in PyTorch and executed on an NVIDIA A100 GPU. We used a composite loss combining Dice loss and binary cross-entropy (BCE). Segmentation quality is measured in terms of specificity, accuracy, Dice, and mIoU.

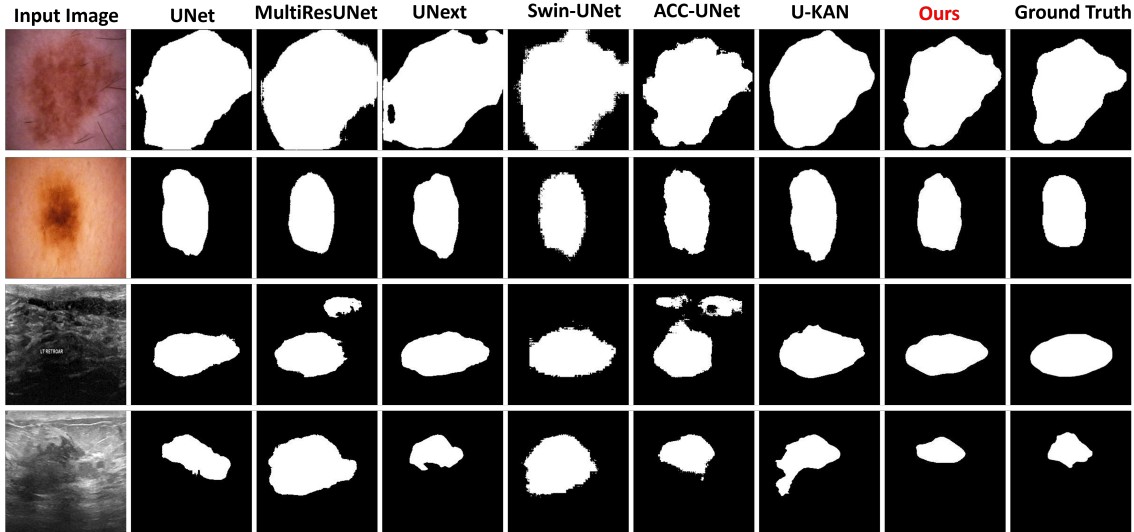

Figure 2: Visual comparison of MedKamba (ours) with state-of-the-art methods: the top two rows show ISIC 2018 results, and the bottom two rows show BUSI results.

## 3.1. Result Analysis

Table 1 and 2 summarize the evaluation of the proposed approach across two benchmark datasets with different imaging modalities, comparing it with various established segmentation approaches. We assessed performance against traditional convolution-based networks, including U-Net (Ronneberger et al., 2015) and MultiRes-UNet (Ibtehaz and Rahman, 2020), as well as the efficient transformer-based model Swin-UNet (Wang and et al., 2022) and recently popular mamba-based VM-Unet(Ruan et al., 2024) and H-vmunet (Wu et al., 2025). In addition, since KAN represents a promising alternative to conventional MLPs, we included comparisons with advanced MLP-style segmentation frameworks like U-NeXt (Valanarasu and Patel, 2022), and KAN based U-KAN (Li et al., 2025). The results indicate that MedKamba (Ours) consistently achieves superior performance across two datasets, outperforming the competing methods in both accuracy and reliability. We also provide a detailed visual comparison across all datasets, as shown in Fig. 2. The results indicate that conventional CNN-based models tend to suffer from over or under-segmentation, reflecting their limited ability to capture global context and accurately distinguish between structures. In comparison, MedKamba demonstrates a noticeable reduction in false positive predictions, highlighting its robustness against noisy outputs. When evaluated against SOTA methods, the proposed approach consistently produces segmentation results with sharper boundaries and more precise structural details. Our proposed architecture is lightweight compared to most methods while achieving higher efficiency. These findings emphasize the model's strength in delivering high-fidelity segmentations while preserving fine-grained anatomical features, validating the effectiveness of integrating state space models and $f$-KANs.

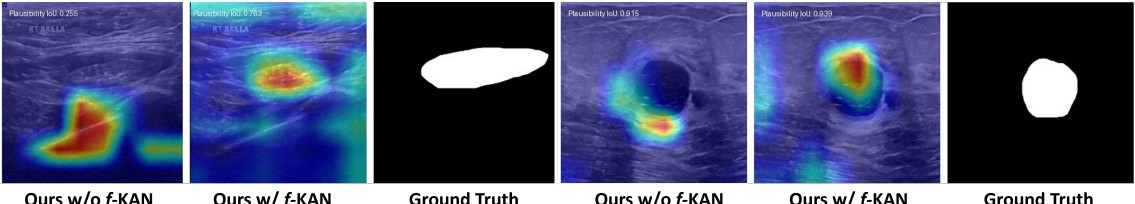

| Ours w/o *f*-KAN | Ours w/ *f*-KAN | Ground Truth | Ours w/o *f*-KAN | Ours w/ *f*-KAN | Ground Truth |

Figure 3: Explainability of MedKamba with channel activation.

### 3.2. Explainability

We additionally investigate the interpretability benefits introduced by the KAN layers by examining their activation behaviors, as illustrated in Fig. 3. When conventional MLP layers are employed (first column), the network exhibits difficulty in generating meaningful activation responses over clinically relevant regions, which is reflected in a low Plausibility IoU score. This metric, proposed in (Cambrin et al., 2024), quantifies the overlap between threshold activation maps and ground-truth masks, with higher values indicating more plausible and reliable explanations. After replacing the MLP layers with fractional KAN layers (second column), a substantial improvement is observed. The activations become more coherent, clearly highlighting target structures and producing boundaries that better match the ground-truth masks (third column). These results indicate that fractional KAN layers improve explainability by aligning activations with anatomically meaningful regions, consistent with prior findings for KAN-based models.

### 3.3. Ablation Study

As shown in Table 3. We conducted an ablation study to analyze the individual contributions of each block and module in MedKamba. Starting from the original U-Net, we use only the first three convolutional layers. The remaining deeper blocks were replaced with our proposed LACE block, resulting in a significant performance improvement. To further enhance performance and refine the skip connections, we incorporated the CAB and SAB modules from MALUNet. These modules enable the model to simultaneously capture global and local contextual information, allowing it to "see" more comprehensively. To increase interpretability and introduce learnable activation functions, we replaced the traditional feedforward network with KAN. Additionally, we experimented with replacing the spline-based activation function (see 4th row in Table 3) with a more flexible Jacobian polynomial activation function (Ours)(see 5th row in Table 3), further improving the model's adaptability and performance.

### 4. Conclusion

This work presents MedKamba, a medical image segmentation framework that blends U-Net's strength in capturing fine-grained local details with enhanced convolutional layers powered by the VSSM-based LACE block to model global context. To improve the quality of skip-connection information, we introduce a KAN-based module, *f*-KSCA. MedKamba is among the first approaches to jointly leverage both Mamba and KAN architectures for

| Configurations | mIoU | Dice |
|---|---|---|
| UNet | 68.30 | 76.80 |
| UNet + LACE | 68.14 | 80.31 |
| UNet + LACE + CAB | 68.93 | 80.82 |
| UNet + LACE + KSCA | 69.63 | 81.22 |
| **UNet + LACE + $f$-KSCA (Ours)** | **71.07** | **82.17** |

Table 3: Ablation study results comparing different network variants on BUSI Dataset.

medical image analysis. Comprehensive experiments across multiple datasets show that MedKamba delivers strong, consistent performance across various modalities and tasks. Ablation studies also highlight the complementary advantages of Mamba and KAN, leading to better optimization efficiency and higher segmentation accuracy. This study focuses on 2D medical image segmentation. Investigating the extension of MedKamba to 3D volumetric datasets, such as CT and MRI, will be explored in future work.

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
