# OpenReview forum: "MedKamba: A Novel Approach Integrating State-Space Models and Fractional Kolmogorov–Arnold Networks for Medical Image Segmentation."
_MIDL.io/2026/Conference — MIDL 2026 Poster_

### Official Review · Reviewer_CCkm · 2026-01-06

**Confidence:** 4
**Preliminary Rating:** 5
**Final Rating:** 5

**Summary:**

The authors introduce a new segmentation model architecture combining Mamba and a UNet like architecture called MedKamba, introducing several new blocks to help overcome limitations.

They apply their network to 2 popular 2d segmentation datasets and show state of the art results, showing decent improvements over the
previous SOTA.
They also perform a simple ablation study of their novel introductions in the architecture.

**Strengths:**

- The authors show state of the art improvements with their novel architecture on relevant datasets that are not easy to improve on.
Regular UNets are still dominant in medical imaging due to their great performance for their computation costs. Even though transformer based networks can outperform UNets, they come at a large computational cost. Mamba promises the benefits of transformers at a better computational costs, but applying them effectively in a segmentation task is still an open problem.
Therefore explorations in such network architectures are very beneficial, especially if they prove to work well.
- The authors also add a little section to show the explainability of their network activations, which is a nice touch.

**Weaknesses:**

- The authors add 3 different novelties to the model. This adds quite some complexity, and even though there is a simple ablation study, it can be hard to figure out how much each element really contributes.

**Detailed Comments:**

See other comments.

**Justification Of Final Rating:**

The reviewer xm2h makes a lot of good points and some things I didn't quite notice. Maybe with that in mind on my scale (also relative to my other papers) I would have originally gone for a 4/5. Still the authors did a good job to address those comments and improved the paper, putting it back on a 5/5. Indeed a full evaluation on a challenging 3d dataset would really make it excellent, so this 5/5 does not mean its perfect, but I would say quite a strong accept.

**Justification Of The Preliminary Rating:**

The exploration of architectures in this domain are very relevant, as there is a need for efficient architectures that incorporate attention mechanisms in a way that actually improves performance. The validation shows great results on common datasets, so this is a relevant paper in the exploration in the space of architectures.

**Questions To Address In The Rebuttal:**

As the added architecture changes are quite complicated (requiring three novel introductions) some discussion on how the authors arrived at these specific contributions would be interesting.

---

> ### Author Response · Authors · 2026-01-24
>
> We sincerely thank the reviewer for the strong endorsement and for highlighting the relevance of efficient attention-based architectures for medical segmentation.
>
> **a) As the added architecture changes are quite complicated (requiring three novel introductions) some discussion on how the authors arrived at these specific contributions would be interesting.**
>
> The proposed components were derived through an iterative analysis of limitations observed when integrating Mamba-style state-space models into medical image segmentation. The design rationale for each block is summarized below:
>
> ***Motivation 1: Loss of local spatial inductive bias in VSSM-based modeling***
>
> Although Mamba/VSSM captures long-range dependencies efficiently, applying it to 2D feature maps requires spatial flattening, which disrupts neighbourhood continuity and degrades boundary precision. To mitigate this, we introduce the Local Aware Channel Enhancement (LACE) block, which augments VSSM with local convolution to restore spatial locality while preserving global context.
>
> ***Motivation 2: Channel redundancy from large hidden-state dimensions***
>
> Long-range dependency modeling often requires many hidden states, resulting in redundant channel activations and weaker discriminative learning. Within LACE, we integrate channel attention and learnable residual scaling to suppress redundancy and recalibrate feature responses efficiently.
>
> ***Motivation 3: Shallow cross-stage fusion in U-Net skip connections***
>
> Standard U-Net skip connections rely on scale-matched concatenation, limiting cross-stage interaction and ignoring global context from deeper encoder levels. We address this by redesigning skip connections with the fractional Kolmogorov–Arnold spatial–channel attention (f-KSCA) module, which aggregates multi-stage encoder features and generates context-aware channel weights.
>
> ***Motivation for Fractional KANs (f-KAN) with FJAF***
>
> Existing MLP-based attention mechanisms (e.g., CAB in MALUNet [1]) suffer from limited interpretability and quadratic complexity. We therefore adopt fractional KANs with Fractional Jacobi Activation Functions (FJAF)[2] within f-KSCA to enable interpretable nonlinear modeling with improved computational efficiency.
> Their contributions are quantified via a structured ablation study *(Table 3, Section 3.2).* The results show that each component contributes complementarily and that their combination forms the most effective MedKamba architecture.
>
> References:
>
> *1] Jiacheng Ruan, Suncheng Xiang, Mingye Xie, Ting Liu, and Yuzhuo Fu. Malunet: A multiattention and light-weight unet for skin lesion segmentation. In 2022 IEEE International Conference on Bioinformatics and Biomedicine (BIBM), pages 1150–1156. IEEE, 2022.*
>
> *2] Aghaei, Alireza Afzal. "fkan: Fractional kolmogorov–arnold networks with trainable jacobi basis functions." Neurocomputing 623 (2025): 129414.*

---

### Official Review · Reviewer_yFLQ · 2026-01-07

**Confidence:** 4
**Preliminary Rating:** 3
**Final Rating:** 4

**Summary:**

The authors propose MedKamba, a novel U-shaped segmentation approach that employs a hybrid encoder with CNNs–LACE block to effectively capture both local and global contextual information, where tradition skip-connections are replaced with Fractional Kolmogorov–Arnold Networks (f -KANs) to generate channel-wise attention weights from features aggregated across multiple stages. The authors evaluate their method on 2 benchmark datasets.

**Strengths:**

- Very nice Figure 1
- Text is well structured and written which makes it easy to follow and understand their method
- The method is evaluated against a set of baselines, which gives a very good overview over the performance
- Against 8 baselines, the proposed method achieves better Accuracy and Dice for both datasets
- I really like the ablation study and the different model combinations, showing which part achieves which influence
- The authors did a thorough and very detailed analysis on the results, I also like the explainability heatmaps

**Weaknesses:**

- The authors evaluate "against state-of-the-art segmentation models" however did not include the SOTA nnU-Net for medical segmentation --> This is crucial to compare against the SOTA architecture for medical segmentation

**Detailed Comments:**

- A lot of equations where used which are not really referenced at later staged, some might be obsolete
- Table 3 and Figure 3 poorly placed one after the other

**Justification Of Final Rating:**

The authors addressed all of my comments in detail, and also added the nnUNet as a baselines which increased the quality of the paper. Because of this, I raised my initial rating from borderline to a weak accept.

**Justification Of The Preliminary Rating:**

The paper is well-written, with clear figures, thorough analysis, and strong experimental evaluation against multiple baselines. However, including nnU-Net, a leading benchmark in medical image segmentation, would strengthen the comparison and better contextualize the claimed improvements.

**Questions To Address In The Rebuttal:**

The authors should include nnU-Net as an additional baseline, as it is widely regarded as a strong benchmark for medical image segmentation.

---

> ### Author Response · Authors · 2026-01-24
>
> We sincerely thank the reviewer for the very positive evaluation of the writing, figures, ablation studies, and explainability analysis.
>
> **a) The authors should include nnU-Net as an additional baseline, as it is widely regarded as a strong benchmark for medical image segmentation.**
>
> In response, nnU-Net has been added as an additional benchmark on both datasets using the official nnU-Net implementation and the recommended dataset-specific configurations. The corresponding results are included in the updates tables *(see section 3.1, Table 1 and Table 2)* in revised manuscript. Also provided in Tables below in comparison with proposed MedKamba. The added experiments show that MedKamba overcome nnU-Net in terms of Dice, IoU, Specificity and Accuracy on both datasets, while remain computationally efficient. For fair comparison we trained all the baselines from scratch with same dataset split.
>
> *Results on BUSI Dataset*
>
> | Method        | Params (M) | FLOPs (G) | Specificity | Accuracy | mIoU  | Dice  |
> |---------------|------------|-----------|-------------|----------|-------|-------|
> | nnUNet        | 19.1       | 412.7     | 98.39       | 96.51    | 70.02 | 79.45 |
> | **MedKamba**  | **5.50**   | **2.16**  | **98.54**   | **96.62**| **71.07** | **82.17** |
>
>
>
> *Results on ISIC 2018 Dataset*
>
> | Method        | Params (M) | FLOPs (G) | Specificity | Accuracy | mIoU  | Dice  |
> |---------------|------------|-----------|-------------|----------|-------|-------|
> | nnUNet        | 19.1       | 412.7       | 94.32      | 92.21    | 77.03 | 86.43 |
> | **MedKamba**      | **5.50**   | **2.16**        | **95.15**      | **93.00**    | **79.63** | **88.24** |
>
>
>
> **b) Presentation Issues**
>
> We revised the manuscript to better reference equations and improve the placement of Table 3 and Figure 3 for better readability.

---

### Official Review · Reviewer_xm2h · 2026-01-08

**Confidence:** 4
**Preliminary Rating:** 3

**Summary:**

The paper proposes MedKamba, a U-shaped medical image segmentation model that combines Visual State-Space Models with a new Local-Aware Channel Enhancement block to recover local spatial details lost by 1D state-space flattening and to suppress redundant channels. It further replaces standard MLPs with fractional KANs to model multi-stage global context.

**Strengths:**

a) It jointly combines state-space models with fractional KAN, introducing novel LACE and f-KSCA blocks that address both global context modeling and local spatial detail preservation. b) MedKamba outperforms CNN, Transformer, Mamba-based, and KAN-based baselines. c) The paper includes ablation studies and explainability analysis showing that fractional KAN activations produce more meaningful responses.

**Weaknesses:**

a) The evaluation is restricted to only two datasets, both 2D and relatively small, with no validation on large-scale, multi-organ, or 3D volumetric datasets (e.g., CT/MRI), which limits claims of generalizability. b) The baselines overlook some strong recent models (e.g., nnU-Net v2 variants, SAM-based medical adapters, or large pre-trained vision models), making the superiority claims less convincing. c) The paper provides no theoretical analysis or controlled experiments isolating why fractional order is better than standard KAN or polynomial bases. d) Results are reported without confidence intervals, multiple runs, or statistical significance testing, which weakens the reliability of the reported gains.

**Detailed Comments:**

Please see the Strengths and Weaknesses sections provided above.

**Justification Of The Preliminary Rating:**

The preliminary rating is based on a review with particular emphasis on the identified weaknesses outlined in the sections above. These weaknesses highlight several critical gaps in the current proposal, including but not limited to deficiencies in methodology, insufficient clarity in the implementation plan, and limited evidence support.

**Questions To Address In The Rebuttal:**

Please see the Weaknesses sections provided above.

---

> ### Author Response · Authors · 2026-01-24
>
> We thank the reviewer for the detailed and constructive comments, and for acknowledging the novelty of our approach together with the accompanying ablation and explainability analyses.
>
> **a) The current evaluation focuses on two representative 2D medical image segmentation datasets.**
>
> We acknowledge that the current evaluation is conducted on two widely used 2D medical segmentation benchmarks. Our primary goal in this work is architectural validation. Specifically, to study how state-space models and fractional KANs can be effectively adapted for dense segmentation. To clarify scope, we explicitly state that MedKamba is validated on 2D settings in this version. Extending the framework to large-scale and 3D volumetric datasets (e.g., CT/MRI) is a natural next step and is discussed as future work in revised manuscript *(see section 4)*.
>
> **b) The baselines overlook some strong recent models (e.g., nnU-Net v2 variants, SAM-based medical adapters, or large pre-trained vision models).**
>
> Following reviewer feedback, we have added nnU-Net, MedSA (Medical SAM Adapter) as additional baselines using the official framework and recommended configurations. The updated results *(see section 3.1, Table 1 and Table 2)*, show that MedKamba outperforms both methods on both datasets while remaining computationally efficient. Also provided in table below for **ISIC 2018**.
> We also clarified the positioning of our method relative to large-scale pretrained foundation models such as MedSAM. Our proposed method is trained from scratch on only 2.5k images unlike MedSAM *(trained on approx. 1.5 million images)* and still attains competitive results, highlighting its effectiveness as a lightweight and data-efficient alternative.
>
> | Method        | Params (M) | FLOPs (G) | Specificity | Accuracy | mIoU  | Dice  |
> |---------------|------------|-----------|-------------|----------|-------|-------|
> | nnUNet | 19.1|412.7|94.32|92.21|77.03|86.43|
> | MedSA  |104.3| 52.2| 91.09 |92.04  | 77.71| 86.36|
> | MedSAM | 600|NA|98.08  | 95.18| 88.50   | 93.79 |
> | **MedKamba**  | 5.50 |2.16| **95.15** | **93.00** | **79.63**  | **88.24** |
>
> **c) The paper provides no theoretical analysis or controlled experiments isolating why fractional order is better than standard KAN or polynomial bases.**
>
> We agree that deeper theoretical analysis is valuable. In this work, our choice of fractional KANs is primarily motivated by empirical and architectural considerations: fractional Jacobi activation functions [1] enable smoother and more flexible nonlinear approximations with improved numerical stability compared to standard KANs or polynomial bases. This advantage is consistently reflected in our ablation studies *(see 4th and 5th row of Table 3)*, and explainability analyses *(see Figure 3)*, where fractional KANs produce more structured and interpretable activations than MLP/KAN. To address this concern more rigorously, we added in the methodology *(see section 2.3)*, that theoretically motivates the use of fractional-order bases, discussing their approximation properties and stability advantages, and explicitly contrasting them with standard polynomial/KAN formulations.
>
> Reference:
>
> 1] *Aghaei, Alireza Afzal. "fkan: Fractional kolmogorov–arnold networks with trainable jacobi basis functions." Neurocomputing 623 (2025): 129414*.
>
> **d) Results are reported without confidence intervals or statistical testing**.
>
> We agree that statistical reporting strengthens empirical claims. However, we follow the *standard dataset splits and evaluation protocols* commonly adopted in prior works on the evaluated benchmarks to ensure fair and consistent comparison. In response to reviewer concern, we additionally conducted *3-fold cross-validation* experiments on the BUSI dataset, and the corresponding results are reported in table here. Due to time and computational constraints, cross-validation was performed on single dataset only. These additional results further support the stability of the reported performance gains.
>
> **Results on BUSI Dataset**
> | Model| Specificity| Accurac| mIoU| Dice |
> |--------------|--------------------|--------------------|-------------------|-------------------|
> | UNet | 97.62 ± 0.56 | 96.10 ± 0.33 | 66.82 ± 1.81| 75.74 ± 1.40 |
> | MultiResUNet | 96.01 ± 0.36 | 94.84 ± 0.27| 64.73 ± 1.90 | 74.19 ± 2.79 |
> | nnUNet |98.23 ± 0.61  | 95.54 ± 0.23 | 68.91 ± 1.04 | 78.21 ± 1.27|
> | UNext |97.82 ± 0.22 | 96.19 ± 0.13| 69.33 ± 0.98| 78.67 ± 1.02|
> | Swin-UNet | 97.12 ± 0.27|96.10 ± 0.21| 69.21 ± 0.72| 77.98 ± 0.81|
> | ACC-UNet  | 95.69 ± 0.96| 95.65 ± 0.83| 69.51 ± 1.29| 78.08 ± 1.01|
> |MedSA| 97.21 ± 0.35 | 94.96± 0.46| 66.19 ± 1.12 |76.90± 1.61|
> | VM-UNet | 97.21 ± 0.39| 94.34 ± 0.93 | 60.21 ± 1.52| 75.32 ± 1.23|
> | U-KAN | 97.22 ± 0.21|95.80 ± 0.17| 68.63 ± 0.70| 79.32 ± 0.40|
> | H-vmUnet | 98.13 ± 0.34| 96.21 ± 0.43| 66.21 ± 0.34| 78.21 ± 1.54|
> | **MedKamba**| **98.56 ± 0.14**   | **96.98 ± 0.12** | **70.01 ± 1.12**  | **81.31 ± 0.83** |

---

### Author Response · Authors · 2026-01-24

We thank the reviewers for their thorough and constructive feedback on our manuscript. We appreciate the positive reception of our work, particularly regarding its novelty in medical imaging and the strength of the experimental evaluation, ablation studies, and explainability analysis. In response to the reviewers’ helpful comments and suggestions, we have incorporated additional baseline comparisons and further refined our analysis throughout the paper. We will release the code upon acceptance and ensure that it is well documented, with dataset usage details made publicly available to support reproducibility and future research. Below, we address the specific concerns raised by each reviewer. All changes to the main manuscript are highlighted in red.

---

### Author Rebuttal · Authors · 2026-01-24

**Rebuttal:**

We sincerely appreciate the reviewers’ constructive feedback, which has helped improve the clarity of our manuscript. Please find the revised manuscript as a PDF with all changes highlighted in red.

**Supporting Material:**

/attachment/b62d02521a45e50a4e935218b481dacb89665d58.pdf

---

### Meta-Review · Area_Chair_N5x2 · 2026-02-08

**Recommendation:** Accept (Poster)
**Confidence:** 3

**Metareview:**

The reviewers suggested that this paper is a borderline, and the authors have posted a rebuttal, which answered the raised questions.

The overall meta-review for this paper is that the work needs to be accepted, as the final scores are 3, 4, 5.

---

### Decision · Program_Chairs · 2026-02-13

Accept (Poster)